A new bilaterally injured trilobite presents insight into attack patterns of Cambrian predators

Zong Ruiwen zongruiwen@cug.edu.cn 1
Bicknell Russell D.C. 2
1 State Key Laboratory of Biogeology and Environmental Geology, China University of Geosciences , Wuhan , China
2 Palaeoscience Research Centre, School of Environmental and Rural Science, University of New England , New South Wales , Australia
De Baets Kenneth
Electronic publication date: 2022 Oct 10
Publication date: 2022
Volume: 10
Electronic Location ID: e14185
Received 2022 Jul 1; Accepted 2022 Sep 14
Copyright: ©2022 Zong and Bicknell
Copyright year: 2022
Copyright holder: Zong and Bicknell
License: This is an open access article distributed under the terms of the Creative Commons Attribution License, which permits unrestricted use, distribution, reproduction and adaptation in any medium and for any purpose provided that it is properly attributed. For attribution, the original author(s), title, publication source (PeerJ) and either DOI or URL of the article must be cited.
License URL: https://creativecommons.org/licenses/by/4.0/

Keywords: Predation, Regeneration, Redlichia, Cambrian, South China

Funding: National Natural Science Foundation of China 42072041 Australian Research Council DP200102005 University of New England Postdoctoral Fellowship This work was supported by the National Natural Science Foundation of China (No. 42072041), an Australian Research Council grant (DP200102005) and a University of New England Postdoctoral Fellowship. The funders had no role in study design, data collection and analysis, decision to publish, or preparation of the manuscript.

==============================
Durophagous predation in the Cambrian is typically recorded as malformed shells and trilobites, with rarer evidence in the form of coprolites and shelly gut contents. Reporting novel evidence for shell-crushing further expands the understanding of where and when in the Cambrian durophagy was present. To expand the current documentation and present new records of malformed trilobites from the Cambrian of China, we present an injured Redlichia (Pteroredlichia) chinensis from the lower Cambrian Balang Formation, western Hunan, South China. The specimen has two distinct injuries along the thorax. The injuries show different degrees of regeneration, suggesting that the specimen was attacked twice. We propose that the individual may have been targeted more readily for the second attack. This predatory approach would have been highly energy efficient, maximizing net energy gain during the attack.

Introduction

Biomineralized trilobite exoskeleton was constructed from two layers of low magnesium calcite (Wilmot & Fallick, 1989; Hughes, 2007). Due to this construction, trilobites had a markedly durable dorsal exoskeleton when compared to most other Paleozoic euarthropods (Hughes, 2007). Although this cuticular construction protected the soft-bodied sections, these exoskeletons were still susceptible to damage from shell crushing (durophagous) predators and boring organisms especially post-molting, during the soft- and paper-shell stages (e.g., Owen, 1985; Babcock, 1993; Pratt, 1998; Zamora et al., 2011; Fatka, Budil & Grigar, 2015; Pates et al., 2017; Bicknell & Paterson, 2018; Bicknell & Pates, 2020; De Baets et al., 2021). This is evidence by the recorded in malformed and injured specimens (Owen, 1985). These injuries provide insight into Paleozoic predation strategies, predator–prey interactions, evolution of trilobite morphology and behavior, and Paleozoic foodwebs (Babcock & Robison, 1989; Babcock, 1993; Babcock, 2003; Babcock, 2007; Nedin, 1999; Brett & Walker, 2002; Brett, 2003; Bicknell & Paterson, 2018; Bicknell et al., 2022a).

While there have been rare instances of accidental injuries (Rudkin, 1985), the majority injured Cambrian and Ordovician trilobites are attributed to failed predation (Owen, 1985; Babcock, 1993; Bicknell & Paterson, 2018; Bicknell et al., 2021; Bicknell et al., 2022a). Within this fossil record, most injured specimens show unilaterally expressed injuries (Babcock, 1993; Bicknell & Paterson, 2018 and their references), with rarer evidence of multiple injuries across the exoskeleton (e.g., Šnajdr, 1979; Conway Morris & Jenkins, 1985; Ou et al., 2009; Pates & Bicknell, 2019; Bicknell & Holland, 2020; Bicknell & Pates, 2020; Bicknell et al., 2021; Bicknell et al., 2022a). Determining whether specimens with multiple injuries record one or multiple failed attacks is complex. However, the extent of exoskeletal regeneration can be considered a proxy for understanding the timing of attacks and subsequent recovery (e.g., Cheng et al., 2019; Zong, 2021a). To expand on this line of enquiry, we describe a malformed redlichiid trilobite from the Cambrian-aged Balang Formation (western Hunan, South China). This specimen shows a bilaterally expressed malformation and is used to explore patterns of trilobite regeneration, presenting new insight into early Cambrian predation strategies.

Materials and Methods

The examined specimen was collected from the Balang Formation, Huayuan County, Xiangxi Autonomous Prefecture, Hunan Province (Fig. 1). The Balang Formation is widely distributed in eastern Guizhou and northwestern Hunan, and is comprised of fine calcareous clastic rocks with limited limestone interbeds or lenses. The formation was therefore likely deposited in a shelf to slope environment (Fig. 1; Yin, 1996; Liang et al., 2017). The formation is located within the Arthricocephalus chauveaui-Changaspis elongata zone, which corresponds to the Cambrian Series 2, Stage 4 (Peng, 2009; Qin et al., 2010). The Balang Formation has yielded a diverse, exceptionally preserved fauna including radiodonts, trilobitomorphs, bivalved arthropods, worms, chancelloriids, cnidarians, echinoderms, and algae (Peng et al., 2005; Liu & Lei, 2013). Trilobites from this formation consist of primarily oryctocephalids, redlichiids, and ptychopariids that are arrayed across ten genera (Peng et al., 2018; Chen, 2019). The Balang redlichiids consist of four species (including one subspecies) within a Redlichia subgenus (Liang et al., 2017; Chen & Zhao, 2018), and the dominate taxon is Redlichia (Pteroredlichia) chinensis (Walcott, 1905). The dark gray calcareous shale of the Balang Formation in western Hunan has yielded a large number of well-preserved R. (Pteroredlichia) chinensis (Zong, 2021b). The malformed individual was assigned to this species and represents an internal mold of R. (Pteroredlichia) chinensis. The examined specimen was coated with magnesium oxide for photography. All photographs were taken with a Nikon D5100 camera using a Micro-Nikkor 55 mm F3.5 lens. Specimen measurements were made with ImageJ software (Schneider, Rasband & Eliceiri, 2012). The specimen is housed in the State Key Laboratory of Biogeology and Environmental Geology, China University of Geosciences (Wuhan) (BGEG).

Figure 1 Geological background map of study area and location of the fossil site.

(A) Cambrian sedimentary facies zones of South China (modified from Zhao et al., 1993). (B) Map of fossil site at Huayuan County, Xiangxi Autonomous Prefecture, Hunan Province. (C) Stratigraphic series showing relative position and age of the Balang Formation (modified from Zhu et al., 2021).

Results

The described specimen (BGEG-HXB-02) is an articulated exoskeleton lacking free cheeks and is therefore likely an exuvia (Daley & Drage, 2016; Drage, 2019). Two malformations along the thorax are noted (Fig. 2A). The more anterior malformation is an asymmetrical V-shaped indentation along the fourth and fifth pleural segments of the left pleural lobe showing limited cicatrization (Fig. 2B). The distal section of the fourth pleural segment is lacking a pleural spine, truncated by ∼1.8 mm. The pleural furrow of this segment is slightly S-shaped. The fifth thoracic segment is truncated by ∼2.9 mm and reduced abaxially. Distal section of this segment is rounded and reduced by ∼50% relative to the undamaged sixth pleural segment. The pleural furrow of this segment is also slightly distorted. Additionally, the third segment on the left side was rotated anteriorly and positioned under the second pleural segment. As this rotated segment lacks shortening or distortion (Fig. 2C), this reflects either taphonomic alteration or displacement during molting (Zong, 2021a).

Figure 2 Injured Redlichia (Pteroredlichia) chinensis (Walcott, 1905) from the Balang Formation (Cambrian Series 2, Stage 4), Hunan, South China.

(A) Complete specimen (BGEG-HXB-02) showing two injuries. (B, C) Close-up of the injury on the left pleural lobe. (B) V-shaped indentation. (C) Same as B showing overlap of the second (yellow) and third (red) pleural segments and injury (blue). (D, E) Close-up of injury on right pleural lobe. (D) U-shaped indentation. (E) Same as D showing injury (blue). Abbreviation: ts., thoracic segment.

The second malformation is a U-shaped indentation located across the seventh to ninth pleural segments on the right pleural lobe (Figs. 2D and 2E). The seventh and eight segments are truncated by ∼1.8 mm and ∼2.6 mm, respectively. The ninth segment is truncated by 2.5 mm, and the distal portion is narrower when compared to the ninth segment on the left pleural lobe. All malformed segments have reduced pleural spines that are ∼50% smaller when compared to undamaged segments, indicating pleural spine regeneration (Pates et al., 2017).

Discussion

Several factors likely produced malformations in trilobites. These include failed predation or molting resulting in injuries, developmental malformations producing teratologies, and pathological infections (e.g., Šnajdr, 1978; Rudkin, 1985; Owen, 1985; Babcock, 1993; Babcock, 2003; Owen & Tilsley, 1996; Chen, 2011; Fatka, Budil & Grigar, 2015; Bicknell & Pates, 2020; Bicknell & Smith, 2021; De Baets et al., 2021; Zong, 2021b). Morphologically comparable abnormalities are observed in modern arthropods (Juanes & Smith, 1995; Brandt, 2002; Pandourski & Evtimova, 2005; Pandourski & Evtimova, 2009; Hopkins & Das, 2015; Bicknell, Pates & Botton, 2018; Bicknell et al., 2022b; Das et al., 2021; De Baets et al., 2021), supporting the hypotheses that trilobites experienced similar malformations as extant species. Abnormal trilobite and horseshoe crab specimens with U-, V- or W-shaped breakages, reduced segments, and evidence of cicatrization or regeneration are considered indicative of non-lethal predation (Šnajdr, 1978; Babcock, 1993; Nedin, 1999; Zhu et al., 2007; Bicknell & Paterson, 2018; Bicknell, Pates & Botton, 2018; Bicknell et al., 2022a; Pates & Bicknell, 2019; Bicknell & Pates, 2020; Zong, 2021b). Two indentations documented here are V- and U-shaped with cicatrization on the left injury and pleural spine regeneration on the right injury. Given this, we confidently assign these indentations to failed predation within the Balang Formation. More importantly, these injuries show distinctly different stages of recovery (Fig. 2).

Trilobites are thought to have recovered from injury in an antero-posterior manner, such that anterior segments show markedly more recovery than posterior sections (e.g., Šnajdr, 1981; Conway Morris & Jenkins, 1985; Babcock, 1993; McNamara & Tuura, 2011). This pattern is common to modern arthropods and annelids and is controlled by segmentation polarity genes (McNamara & Tuura, 2011). In BGEG-HXB-02, the anterior injury shows cicatrization, but no evidence for segment regeneration. This presents an apparent conundrum. If we assume the injuries were incurred at the same time, we directly contradict fundamental theories on arthropod development (McNamara & Tuura, 2011). The most parsimonious explanation for the observed pattern is that the posterior injury was incurred first. This injury was able to regenerate before the more anterior injury was incurred.

The presence of two injuries from two distinct attacks demonstrates that Cambrian trilobites could have experienced multiple attacks during their life cycle. This has important implications for Cambrian predator–prey systems, especially with comparison to modern systems. Extant predators in both terrestrial and marine ecosystems will target weaker prey within a population as these individuals require less energy in predation and likely represent a more successful attack (Temple, 1987; Mesa et al., 1994; Hethcote et al., 2004; Peharda & Morton, 2006; Genovart et al., 2010). It seems that predation targeting more vulnerable individuals had therefore arisen in the Cambrian and may have allowed the first durophages to maximizing net energy gain (i.e., energy derived from prey after accounting for energy lost during predation; Forsman, 1996; Gosselin & Chia, 1996) during predation. Finally, the rarity of trilobite specimens with multiple distinct injuries likely reflects an increased rate of successful predation, and a higher rate of mortality in previously injured individuals.

We are grateful for the constructive comments from Kenneth McNamara, Jean Vannier, Oldřich Fatka and editor Kenneth De Baets that improved the clarity and scope of the manuscript.

Additional Information and Declarations

Competing Interests

Author Contributions

Data Availability

The authors declare there are no competing interests.

Ruiwen Zong conceived and designed the experiments, performed the experiments, analyzed the data, prepared figures and/or tables, authored or reviewed drafts of the article, and approved the final draft.

Russell D.C. Bicknell performed the experiments, analyzed the data, authored or reviewed drafts of the article, and approved the final draft.

The following information was supplied regarding data availability:

The raw data are available in Figures 1 and 2.

The specimen is stored in the State Key Laboratory of Biogeology and Environmental Geology, China University of Geosciences, Wuhan, China: BGEG-HXB-02.

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
