# Peer review of "A new bilaterally injured trilobite presents insight into attack patterns of Cambrian predators"

_PeerJ, doi:10.7717/peerj.14185_

## Round 0.1 · original submission · Minor Revisions

You provide a detailed account of a trilobite specimen which was plausibly attacked by two separate non-lethal attacks. In doing this, your manuscript provides a valuable contribution to the overall understanding of predations in early metazoans. I would love to see this published, but there are some points which needs to be addressed before publication.

Background: I agree with reviewer 2 that the manuscript needs additional background information on vulnerability, molting (regeneration during or between molting: compare Jell 1989), healing and potential causes – particularly those not so familiar with trilobites or arthropods more generally (see particularly suggestions by reviewer 1)

Comparative studies in modern marine arthropods: Additional literature and discussion on similar structures and their causes in modern forms or comparative studies would be crucial to support your interpretations (e.g., Jell 1989). This is not necessarily straightforward – I was also asked to do something similar in a manuscript (e.g., De Baets et al. 2021) and took some while to acquire the literature but our arguments and contribution became stronger after it. For this aspect and identifying potential mechanisms/culprits, it would be crucial to state if you only investigated the internal mold or also the external one (compare reviewer 3)

Multiple attacks: the interpretation of multiple attacks is an interesting phenomenon and might be worth elaborating on – particularly on its rarity and potential changes related with changes in predation over time (see reviewer 2)


Durability of skeleton and potential culprits: some additional arguments and discussion for the durability of the trilobite skeleton and the potential culprits would also be crucial (compare reviewer 2)

Typographical/Formatting issues: reviewer 1 pointed out some grammatical points.

Cited literature: Apart from some additional reference on modern arthropods, some additional work on fossil forms might be worth citing and/or discussing (see reviewer 3)

Please make sure to address these and all suggestions by the reviewers including those listed in annotated pdfs.

I look forward to receiving the revised manuscript.

·

Basic reporting

The article is clearly written. There are a few grammatical points that the authors need attending to which I append below. The introduction and background to the study are clearly enunciated and the relevant literature is very well referenced. The single figure is very clear and shows the points discussed in the text very clearly.

Experimental design

The research carried out by the authors is clearly within the scope of the journal. Their aims are well defined and the authors clearly state how the description and interpretation of this specimen contributes to the overall understanding of the impact of durophagy in early metazoans. The methods are well described.

Validity of the findings

The authors provide a good, detailed account of a trilobite that they claim has suffered two separate non-lethal attacks. Their arguments for this are solid. As such they make a useful contribution to the literature on predation pressure in early metazoan animals.

Additional comments

I recommend that the paper be accepted for publication with the proviso that they attend to the minor grammatical points that I append below.

Line 13, replace 'durophage' by 'durophagy'
Line 26, comma, not semicolon
Line 40, delete comma
Line 49, Should read '...and is comprised...
Line 50, 'lenses'
Line 61, '...has yielded...'
Line 61, ...'the malformed...'
Lines 62, 63, delete sentence starting 'The abnormal specimen...'
Line 70, should be 'exuvia' as only one

·

Basic reporting

Line 25-26- Trilobites and other arthropods were especially vulnerable just after molting and before they completely renewed their calcified cuticle. Perhaps you could mention this ?

Line 39- How can we be sure that these injuries are due to attacks and not to impacts against a hard material ? In contrast with the cephalon and pygidium that are thick and have a rounded shape, the thorax consists of potentially far more fragile elements. Please add a few sentences somewhere in the MS.

Line 70- exuvia (singular). What do you mean here ? The trilobite was attacked, survived the attack by cicatrazing the wound, molted its old (injured) cuticle and made a new one with fully regenerated thoracic elements ? To me regeneration occurs after the animal molts and can be more or less successful. Please make this chronology clearer for the reader. Please, use examples from extant crustaceans (add relevant refs). Exuviae are lighter than whole animals and can be easily transported by currents which increases their likelihood of being broken.

Line 24- You mean “when compared ?”. Why do you think trilobites had a more durable dorsal exoskeleton. It is not clear. What about other arthropods (e.g. Fuxianhuia, Sidneyia)

Line 26- and also non-biological physical damages

Description – The figured trilobite clearly shows anomalies on both sides of the thorax. The tips of the thoracic elements (right side) seem to have cicatrized. I can hardly see traces of regeneration. Please give details here. Again check in the litterature when and how cicatrization occurs in extant crustaceans. Concerning the left side ts 2 and 3 are probably simply displaced.

Line 84- pleural

Line 89- again, please provide supporting evidence from extant crustaceans

Line 96-97- Why would it be different ? (regeneration on right side; cicatrization only left side). It sounds a bit odd. Please explain. I am certain that trilobites were able to heal wounds (e.g. by clotting hemolymph) but less sure that regeneration occurred during the intermolt stage. This has to checked by looking modern crustaceans.

Line 100- This sentence is unclear. I don’t understand.

Line 110. Your idea of multiple attacks is interesting. In the present case, both attacks during a very short time (shorter than the whole animal’s life) between two molting events. It is possible that the predator made a first attempt then a second one immediately after while struggling with its prey. To me this would be the most plausible interpretation.

Line 112- It is true with lions but I don’t think marine invertebrates can spot a “weaker” or “vulnerable” individuals among populations. More likely they successfully attack isolated or slow-moving specimens they meet by chance.

Line 117- What do you mean by « maximizing net energy gain » ? Please explain in more details.

Line 118- Finally, the rarity of trilobite specimens with multiple distinct injuries likely reflects an increased rate of successful predation, and a higher rate of mortality in previously injured individuals.
It is true that trilobites with injuries are rare. These survived non-lethal attacks from predators or other non-biological physical impacts. It does not imply that many trilobites died from successful attacks (no fossil evidence). To me, large trilobites were rarely attacked and/or could easily escape from predators because of their muscle power. In contrast, juveniles were more easily preyed upon. The diet of Sidneyia consists of larval trilobites (see Zacai, Vannier et al.), especially one particular species. The adults of this species are abundant in the Burgess Shale. No injuries due to possible attacks have been reported in these adult specimens. Please, improve your conclusions.

What would be the potential predator of your trilobite

Jean VANNIER

Experimental design

no experiments were conducted

Validity of the findings

see remarks above

Additional comments

none

·

Basic reporting

The quality of English is ensured by native speaker R. Bicknell

Literature references are relevant and are appropriately referenced.
(I did one correction and proposed to cite two more contributions, which are marked also in the text).

The article structure and both figures are relevant appropriately described and labelled and submitted in acceptable format.

The contribution represents an appropriate ‘unit of publication’, and includes all results relevant to the hypothesis.

Experimental design

The contribution represents a primary research and fits with Aims and Scope of peerJ.
The relevant research question is well defined; it is answered after rigorous and adequate investigation in conclusions.
Proposed suggestions are written directly in the pdf

Validity of the findings

The contribution describes extraordinary trilobite specimen and logically explains the origin of observed anomalous development of the exoskeleton.

Conclusions are well stated, linked to original research question & limited to supporting results.

Additional comments

The contribution is of high quality and recommend it to be accepted after correction of proposed improvements.
Summary: minor revision.

As i do not see remarks made in the pdf, here is a summary of all proposed changes.
Line 24
composed or compared
Line 26
The contribution is of high quality and recommend it to be accepted after correction of proposed improvements.
Summary: minor revision.
Line 38
... 1985; Ou ...
add here at least Šnajdr, 1979
Šnajdr, M., 1979. Two trinucleid trilobites with repair of traumatic injury. Věstník Ústředního ústavu geologického 54 (1), 49–50.
Line 63
You studied only the internal mould, or also the externam one.
This is to be explicitly stated here.
Line 65
et al.
Line 70
et al.
Line 90
Here, the contribution of Šnajdr (1978) should be cited, as such malformation was described in numerous specimens.
Šnajdr, M., 1978. Anomalous carapaces of Bohemian paradoxid trilobites. Sborník geologických věd, Paleontologie 20, 7–31.
Line 91
Here, the contribution of Šnajdr (1978) should be cited, as such malformation was described in numerous specimens.
Šnajdr, M., 1978. Anomalous carapaces of Bohemian paradoxid trilobites. Sborník geologických věd, Paleontologie 20, 7–31.
Line 94
Here, the contribution of Šnajdr (1978) should be cited, as such malformation was described in numerous specimens.
Šnajdr, M., 1978. Anomalous carapaces of Bohemian paradoxid trilobites. Sborník geologických věd, Paleontologie 20, 7–31.
Line 145
Bruthansová
Line 235
add here
Šnajdr, M., 1978. Anomalous carapaces of Bohemian paradoxid trilobites. Sborník geologických věd, Paleontologie 20, 7–31.
Šnajdr, M., 1979. Two trinucleid trilobites with repair of traumatic injury. Věstník Ústředního ústavu geologického 54 (1), 49–50.

---

## Round 0.2 · accepted · Accept

Thank you for adequately addressing all suggestions. The manuscript has become even more comprehensive and easier to follow. The introduction of additional comparison with modern taxa as well as a definition of net energy gain are also greatly appreciated. For future reference, please make sure when adding the definition of net energy gain in text to also add it consistently and completely to the rebuttal. I only noticed some very minor formatting issues (De Baets et al. 2022b should likely be De Baets et al. 2021 throughout; please check all references one more time for consistency), which can and should be resolved during the proofing phase. I check the manuscript in detail and i feel your manuscript can be accepted without further review.